# Oscillating light engine realized by photothermal solvent evaporation

Jingjing Li[1,2], Linlin Mou[3], Zunfeng Liu [3] ✉, Xiang Zhou [1] ✉ & Yongsheng Chen [3] ✉

Continuous mechanical work output can be generated by using combustion engines and electric motors, as well as actuators, through on/off control via external stimuli. Solar energy has been used to generate electricity and heat in human daily life; however, the direct conversion of solar energy to continuous mechanical work has not been realized. In this work, a solar engine is developed using an oscillating actuator, which is realized through an alternating volume decrease of each side of a polypropylene/carbon black polymer film induced by photothermal-derived solvent evaporation. The anisotropic solvent evaporation and fast gradient diffusion in the polymer film sustains oscillating bending actuation under the illumination of divergent light. This light-driven oscillator shows excellent oscillation performance, excellent loading capability, and high energy conversion efficiency, and it can never stop with solvent supply. The oscillator can cyclically lift up a load and output work, exhibiting a maximum specific work of $30.9 \times 10^{-5}$ J g$^{-1}$ and a maximum specific power of $15.4 \times 10^{-5}$ W g$^{-1}$ under infrared light. This work can inspire the development of autonomous devices and provide a design strategy for solar engines.

Plants harvest solar energy via photosynthesis to maintain various biological processes, such as propagation and growth[1]; the energy of the natural climate and meteorological phenomena (such as wind and rain) and other motions originates from the photothermal conversion of solar energy[2,3]. Inspired by these natural processes, energy conversion from solar to electric, chemical and thermal energy is of great significance for human daily life and industry[4–8]. Mechanical energy is also widely needed in human daily life and can be continuously generated from combustion engines by burning fossil fuel or from electric motors. Direct conversion of solar energy into mechanical energy is highly desired, which can avoid efficiency loss in the multiple steps of energy conversion, storage, and delivery. To date, a solar engine that can directly convert solar energy into continuous mechanical work output has few been developed.

Actuators can generate bending, contraction, elongation, and rotation morphing motions upon exposure to external stimuli to output mechanical energy[9–12]. Different types of actuators have been successfully developed, including dielectric elastomer actuators[13,14], conducting polymers[15,16], shape memory polymers and alloys[17,18], twisted fiber artificial muscles[19,20], ionic polymer composites[21], pneumatic actuators[22], and other inorganic materials[23,24]. The major application area of thin film actuator is either actuation or morphing. Actuator usually needs large force to do a certain function while the morphing usually needs smaller force just to deform itself[25]. Because morphing application usually needs smaller force and energy, it has a relatively thinner film thickness compared with actuation application. In this work, in order to obtain output work, a high actuation stress, a large morphing amplitude, as well as a high actuation frequency, are highly desired. To realize continuous work output of an actuator, a

[1]Department of Science, China Pharmaceutical University, Nanjing 211198, China. [2]College of Chemistry and Chemical Engineering, Anyang Normal University, Anyang, Henan 455000, China. [3]State Key Laboratory of Medicinal Chemical Biology, Key Laboratory of Functional Polymer Materials, Frontiers Science Center for New Organic Matter, College of Chemistry, Nankai University, Tianjin 300071, China. ✉e-mail: liuzunfeng@nankai.edu.cn; zhouxiang@cpu.edu.cn; yschen99@nankai.edu.cn

pulsed stimulus is generally required by periodically switching on/off the supply of moisture[26,27], electricity[28,29], light[30-34], etc., or by using an alternating current electrical supply. However, alternatively switching on/off sunlight to realize oscillating actuation is not convenient.

Recently, there have been a few successful advances in oscillating bending actuation under light irradiation[35,36]. A general strategy is to confine the incident light in a localized region (e.g., laser beam) so that the light-induced bending causes the actuator to be no longer illuminated by the light beam and bend back to be illuminated again[37-45]. This requires the incident light to have a very strict beam width and a special irradiation angle. As sunlight is characterized as diffuse light, which does not have a strict beam width and the incident angle varies with time, the actuator has difficulty escaping from sunlight irradiation during actuation. Currently, preparing an oscillating actuator using diffuse sunlight or divergent light that can work as an engine to generate oscillating bending actuation is still a challenge.

For oscillating bending under divergent light, the actuator should meet the following requirements. (1) The actuator should have a symmetric structure, and both film surfaces can shrink under light illumination, which causes bending actuation towards the light and exposure of the other side of the film to the light. (2) The decreased volume should quickly recover when there is no light illumination. The time for volume recovery and shrinkage would match so that the volume can be fully recovered before the next actuation cycle. (3) To obtain efficient conversion from solar to mechanical energy, the oscillating actuation performance (e.g., bending amplitude, frequency, and output work) needs to be optimized. This requires the delicate design of a synergistic combination of actuator mechanical properties (e.g., modules, elasticity, and size), light absorption and conversion, heating and cooling rates, and actuation stress.

Solvent evaporation from a polymer film causes volume shrinkage, and light irradiation on one side of the film accelerates the solvent evaporation, causing anisotropic volume shrinkage and bending actuation towards the light[46,47]. Introducing a porous structure into the film would facilitate the mass transport of solvent molecules, causing a fast bending speed and a large bending amplitude[46,48]. Therefore, light-induced solvent evaporation from a film might be a good candidate to realize oscillating actuation. Until now, there has been no report about oscillating actuators based on the solvent evaporation mechanism.

In this work, we realized oscillating actuation under divergent light, including sunlight, infrared light, and simulated sunlight. When a vertically positioned polypropylene (PP)/carbon black (CB) porous film with continuous solvent supply was irradiated with light, the initial deflection of the film from the central point caused nonequal solvent evaporation from the two sides and asymmetric volume shrinkage. This phenomenon resulted in the actuator bending towards the light and exposing the other side of the film to the light. The fast solvent diffusion in the actuator film caused fast recovery of the decreased volume during unbending of the actuator (Fig. 1a). This oscillating actuation would not stop with continuous supply of solvent and light irradiation and could be realized for different tilt angles of the actuating film and different light irradiation angles through photo-thermal solvent evaporation. The oscillation displacement, thickness-normalized bending curvature and bending curvature at a smaller temperature change during actuation are comparable to other oscillating actuators (Fig. 1b–d)[37-39,49-53]. A solar engine that can continuously output mechanical work was realized through such a design (Fig. 1a).

## Results

### Fabrication and solvent wetting, evaporation, and wicking properties of the oscillating actuator

An ultrathin and high-recovery PP/CB film (Nitto, SCF serials) with acrylic resin binder was used as the oscillator, which was composed of 99.7% polymer (PP, poly(ethyl acrylate)) and 0.3% additives (Ca, Ti). In general, the role of the inorganic additives such as $CaCO_3$ and $TiO_2$ is mainly to enhance the mechanical properties of the PP film (tensile strength, elongation at break and tensile modulus), and to improve the

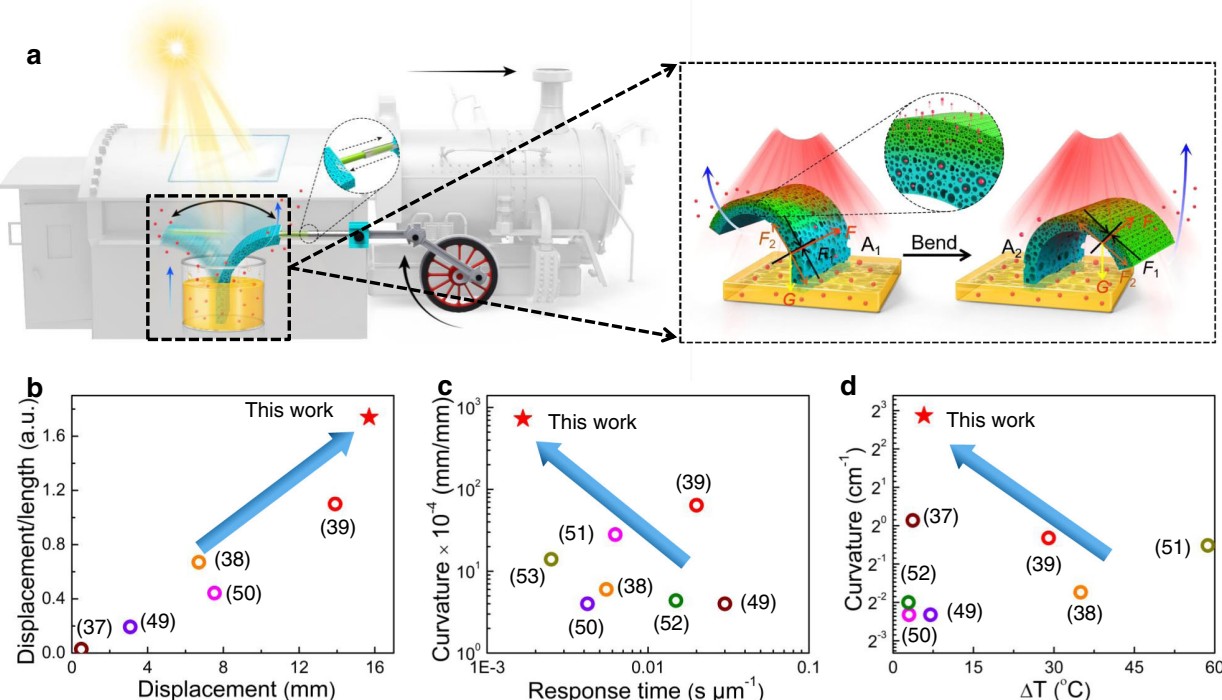

**Fig. 1 | Oscillating Light Engine of the PP/CB actuator. a** Schematic diagram of film as a light motor (the enlarged image is a schematic illustration of the oscillating bending actuation and mechanical analysis). **b** Comparison of the displacement and displacement/length ratio with those of oscillating actuators. **c** Comparison of the curvature and response time normalized by the thickness of the actuator with those of oscillating actuators. **d** Comparison of the curvature and temperature change during actuation of the actuator with those of oscillating actuators[37-39,49-53].

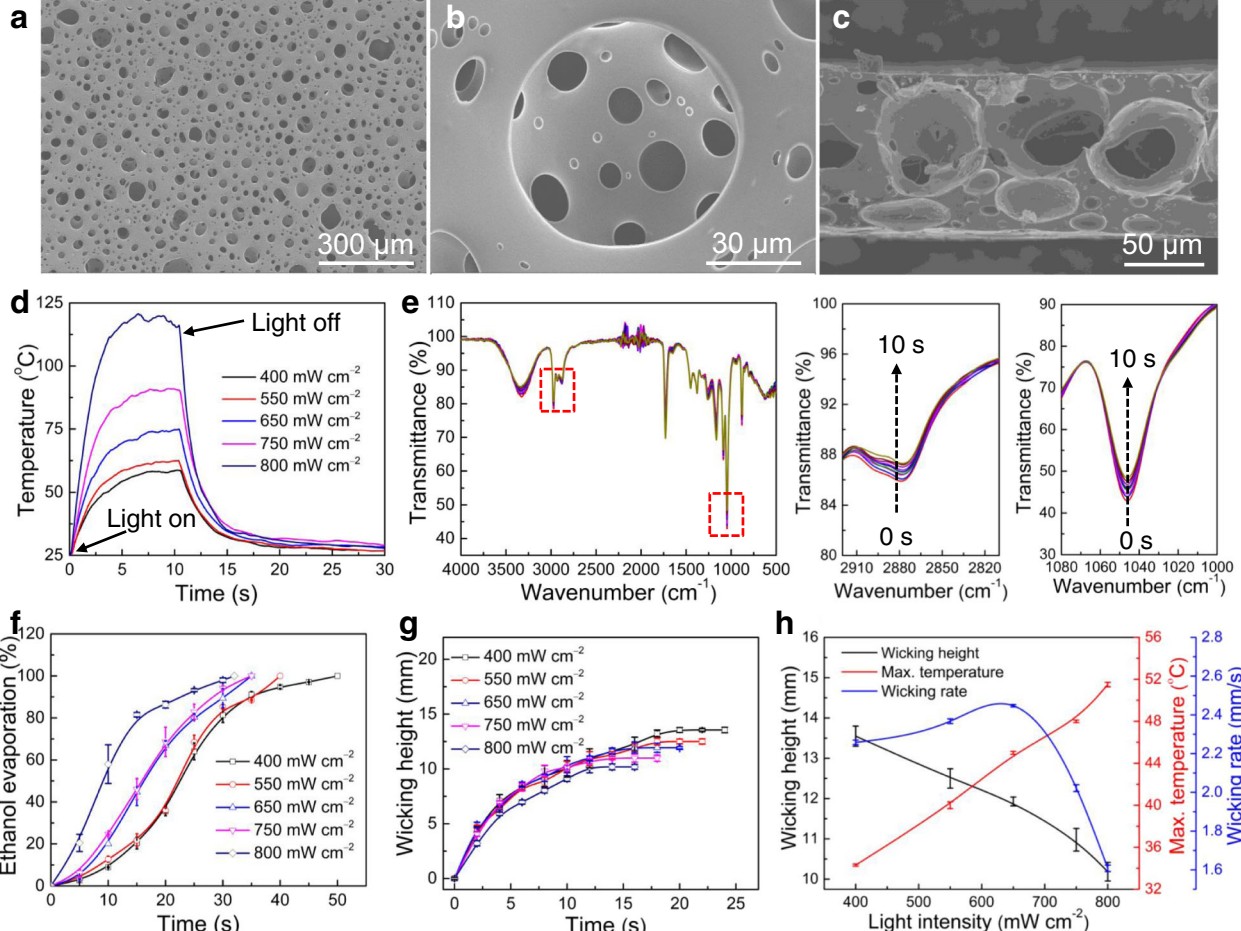

**Fig. 2 | Characterizations of the PP/CB porous film. a, b** Top view and **c** side view of the porous PP/CB film used for the oscillating actuator. **d** Maximum surface temperature as a function of time for the dry porous PP/CB film obtained by switching infrared light with different light intensities on and off. **e** Fourier transform infrared (FTIR) spectra of the porous PP/CB film infiltrated with ethanol exposed to air for different times. The panels to the right show magnified images along the x-axis. **f** Mass percent of evaporated ethanol for the porous PP/CB film infiltrated with ethanol as a function of time under 200 mW cm$^{-2}$ infrared light. **g** Wicking height as a function of time for the porous PP/CB film with ethanol supply under illumination of infrared light with different light intensities. **h** Wicking height, wicking rate, and maximum temperature as a function of light intensity for the porous PP/CB film with ethanol supply. Error bars denote the standard deviation.

foam structure (cell size, cell uniformity, and cell density)[54,55]. First, film with a specific size was vertically mounted into a sponge that can absorb and store ethanol solvents, and the device was then placed in an ethanol reservoir. The film first slightly bent and subsequently generated stable periodic oscillating actuation upon light irradiation as the upper and lower surfaces of the film were alternately irradiated.

The 100 μm-thick film investigated here showed a hierarchical porous structure with microscale average pore size of 40 μm, measured by the mercury intrusion method, and a nanoscale average pore size of approximately 91.83 nm, measured by the N$_2$ adsorption/desorption method (Fig. 2a–c and Supplementary Fig. 1). The morphology of the film with thicknesses of 60 and 150 μm was also obtained (Supplementary Fig. 2). The addition of CB into the porous film resulted in excellent light absorption capability from 200 to 2600 nm, which covers the ultraviolet, visible, and infrared bands (Supplementary Fig. 3). Illumination of the porous PP/CB film by infrared light resulted in a fast temperature increase (Fig. 2d), which could be ascribed to the photothermal effect of CB. An increase in the light intensity caused higher heating speed and film temperature. For example, the film temperature rose to 116 °C in over 10 s under illumination by 800 mW cm$^{-2}$ infrared light, and the temperature decreased to about 35 °C in 5 s when the light was switched off. This light-induced fast heating and cooling speed of the film provides the possibility for use as a photothermal actuator.

The porous PP/CB film coated with polyacrylic adhesive exhibited a hydrophilic surface, which could be wetted by polar solvents (e.g., ethanol). As shown in Supplementary Fig. 4, a drop of ethanol was quickly absorbed by the porous PP/CB surface and formed a contact angle of 10° at 1 s. The ethanol fully infiltrated the pores of the film, as can be seen from the bright regions in the optical microscope (Supplementary Fig. 5), which overlapped with the dark regions observed under the fluorescence microscope, for the porous PP/CB film infiltrated with rhodamine/ethanol solution (0.5 mg mL$^{-1}$). No fluorescence was observed for the pores, which should be because ethanol was absorbed in the pore walls (Supplementary Fig. 6). Absorption of ethanol into the porous PP/CB film resulted in fast swelling of the film. Anisotropic expansion of the length by 18.21%, width by 19.69%, and thickness by 48.50% to a saturated state within 3 s or 5 s was observed for the porous PP/CB film with a size of 3 cm × 1 cm × 100 μm, corresponding to a volume increase of 110.10% (Supplementary Fig. 7). Increasing the film thickness slightly decreased the length increase for ethanol absorption, while it insignificantly affected the changes in the width and thickness of the film. Porous PP/CB films with different thicknesses can be extended by as much as 120% (Supplementary Fig. 8A), and infiltration of ethanol into the porous PP/CB film slightly decreased the breaking strength and modulus while negligibly affecting the fracture strain (Supplementary Fig. 8B), indicating good retention of the mechanical properties after ethanol infiltration. Such

stable mechanical properties of the porous PP/CB film after ethanol infiltration would allow stable actuation performance based on solvent evaporation. An excellent elasticity was obtained during progressive stretching/release from 20 to 80% strain (Supplementary Fig. 9A) and during cyclic stretching/release to 80% strain (Supplementary Fig. 9B).

As solvent evaporation is a key factor affecting the actuation performance of the film, we then investigated the factors affecting ethanol evaporation for the porous PP/CB film, including the thickness of the film and the photothermal effect of CB. We first investigated ethanol evaporation without infrared light illumination. Infiltration of the porous PP/CB film with ethanol resulted in increased characteristic peaks of ethanol at 3328 cm$^{-1}$ (O–H stretching vibration), 2972 cm$^{-1}$ (C–H stretching vibration), and 1046 cm$^{-1}$ (C–O stretching vibration) (Supplementary Fig. 10)[56,57]. Exposing the ethanol-infiltrated porous PP/CB film to open air resulted in ethanol evaporation, as seen from the decreased intensity of the characteristic peaks of ethanol (Fig. 2e). The ethanol content decreased by approximately 50% over 2 min in open air for a porous PP/CB film (3 cm × 1 cm × 100 μm, fully infiltrated with ethanol). Ethanol evaporation can be enhanced by decreasing the film thickness and illuminating the film with infrared light (Fig. 2f and Supplementary Fig. 11). The average ethanol evaporation speed for the 100 μm film increased from 0.53 to 0.89%/s as the light intensity increased from 400 to 800 W cm$^{-2}$ (Supplementary Fig. 12).

The excellent wetting capability of the porous PP/CB film by ethanol encouraged us to investigate whether ethanol can be used as the continuous solvent supply for evaporation-based actuation. A vertically positioned PP/CB film (5 cm × 1 cm × 100 μm) was slowly moved down until it touched the surface of ethanol, and the ethanol was quickly absorbed into the film and climbed to a maximum wicking height of 17.3 mm. A thicker film resulted in an increased wicking height and a faster wicking speed (wicking height per second for wicking of ethanol) (Supplementary Fig. 13), which might be attributed to the higher flow rate and number of pores[58,59]. This wicking height should be determined by the balance between the wetting of the film by ethanol, the absorption of ethanol by the film and ethanol evaporation from the film. Increasing the film thickness increased the ethanol supplied by wetting, while the surface area was negligibly affected, which is highly related to ethanol evaporation.

As infrared light illumination sped up solvent evaporation, we then investigated the wicking height under infrared light with different light intensities. As the light intensity increased from 400 to 800 mW cm$^{-2}$, the wicking height decreased from 13.55 to 10.19 mm (Fig. 2g), indicating a higher ethanol evaporation capacity. As the film was vertically positioned, the temperature along the length of the film infiltrated with ethanol decreased from top to bottom (Supplementary Fig. 14A), possibly due to the increased distance to the light source as well as the gradiently increased ethanol content from the film top to bottom. The temperature at the edge of the ethanol infiltrated into the film also increased with increasing light intensity, and the film temperatures were much lower than those of dried films, as shown in Supplementary Fig. 14B. Interestingly, the wicking rate of ethanol (the advancing height of ethanol per second) increased from 2.26 to a maximum value of 2.45 mm s$^{-1}$ as the light intensity increased from 400 to 650 mW cm$^{-2}$ and then decreased to 1.61 mm s$^{-1}$ as the light intensity further increased to 800 mW cm$^{-2}$ (Fig. 2h). The wicking rate determines how fast the supply of ethanol is for cyclic actuation.

## Oscillating actuator realized by photothermal solvent evaporation

The solvent evaporation-induced volume shrinkage of the ethanol-infiltrated porous PP/CB film under infrared light encouraged us to investigate whether bending actuation would occur under light illumination. A porous PP/CB film (9 mm × 3 mm × 100 μm) infiltrated with ethanol was horizontally placed with one end tethered and the other end free to move. When 800 mW cm$^{-2}$ infrared light was illuminated on

the top surface of the film at a tilt angle of 90° with respect to the horizontal line (vertical to the film surface), the film bent towards the light source, reaching a maximum bending curvature of 8.36 cm$^{-1}$ and exposing the other side to the infrared light. Interestingly, the film unbent back and recovered to the flat shape without switching off the light (Supplementary Fig. 15). The above bending and unbending actuation cycle occurred with damping a few times and finally stopped when almost no ethanol remained in the film (Supplementary Fig. 16). When another drop of ethanol was applied to the film, the above bending and unbending actuation cycle occurred again. This indicated that the ethanol supply highly affected the oscillating bending actuation. We then provided a continuous supply of ethanol by contacting one end of the horizontally placed actuator film to ethanol, with the other end free to move. As a result, continuous oscillating bending actuation with a stable bending curvature (1.81 cm$^{-1}$) was observed by switching on infrared light (Supplementary Fig. 17). This curvature was smaller than that of the first bending actuation cycle without continuous ethanol supply (8.36 cm$^{-1}$), indicating that sucking of ethanol into the porous film decreased the volume difference between the two sides of the film. The bending displacement, i.e., the distance between the free ends of the actuator at the two maximum bending curvatures, reached 5.67 mm, which is also smaller than that without continuous ethanol supply (15.26 mm). This can also be confirmed by the fact that continuous supply of ethanol also decreased the temperature variation during bending actuation compared to the film without continuous ethanol supply.

We then investigated the general applicability of the oscillating actuation for different light irradiation angles and for the actuator film at different tilt angles with respect to the horizontal line. The above horizontally placed actuator film with ethanol supply was irradiated with 800 mW cm$^{-2}$ infrared light at irradiation angles of 45° and 0° (parallel to the film) with respect to the film surface. Interestingly, continuous oscillating actuation was observed for both cases, with a bending curvature of 1.0 cm$^{-1}$ and a bending displacement of 3.2 mm for the 45° tilt angle and a bending curvature of 0.81 cm$^{-1}$ and a bending displacement of 1 mm for the 0° tilt angle (Supplementary Fig. 18). We further investigated the dependence of the actuation performance of the vertically placed actuator film (with a 90° tilt angle with respect to the horizontal line) on the light irradiation angle. Stable oscillating actuation was also observed when the vertically placed actuating film was irradiated with 800 mW cm$^{-2}$ infrared light at irradiation angles of 0°, 45°, and 90°, with bending displacements of 15.7, 17.7, and 17.4 mm, respectively (Supplementary Fig. 19). Further experiments showed that stable oscillation could also be obtained for the actuator film with a 45° tilt angle at different light irradiation angles (0°, 45°, and 90°) (Supplementary Fig. 20). Overall, the above results indicated that oscillating actuation could be realized for different tilt angles of the actuator film and different light irradiation angles through photothermal solvent evaporation.

We studied the detailed characteristics of the oscillating actuation to understand the mechanism using a vertically positioned actuator film irradiated by vertical infrared light (0° tilt angle), which was used in the experiments in the following sections if not otherwise specified. The actuation displacement and frequency and temperature during actuation for the actuator were studied at different light intensities. The oscillating actuation is at a resonance equilibrium state. By switching on the light irradiation, there is a time period for the actuator starting from a static state to reach such a resonance equilibrium state (Fig. 3a and Supplementary Movie 1). Such a time period can be shortened by optimizing the auction conditions, such as by changing film properties, light intensity, solvent supply, etc. For example, by decreasing the ethanol supply from 7 to 1.7 g, such waiting time decreased from 17 to 1.5 s (Supplementary Fig. 21). In our future work, this point would be investigated in detail. Figure 3b shows a series of images for one oscillation cycle of the PP/CB actuator (9 mm × 3 mm

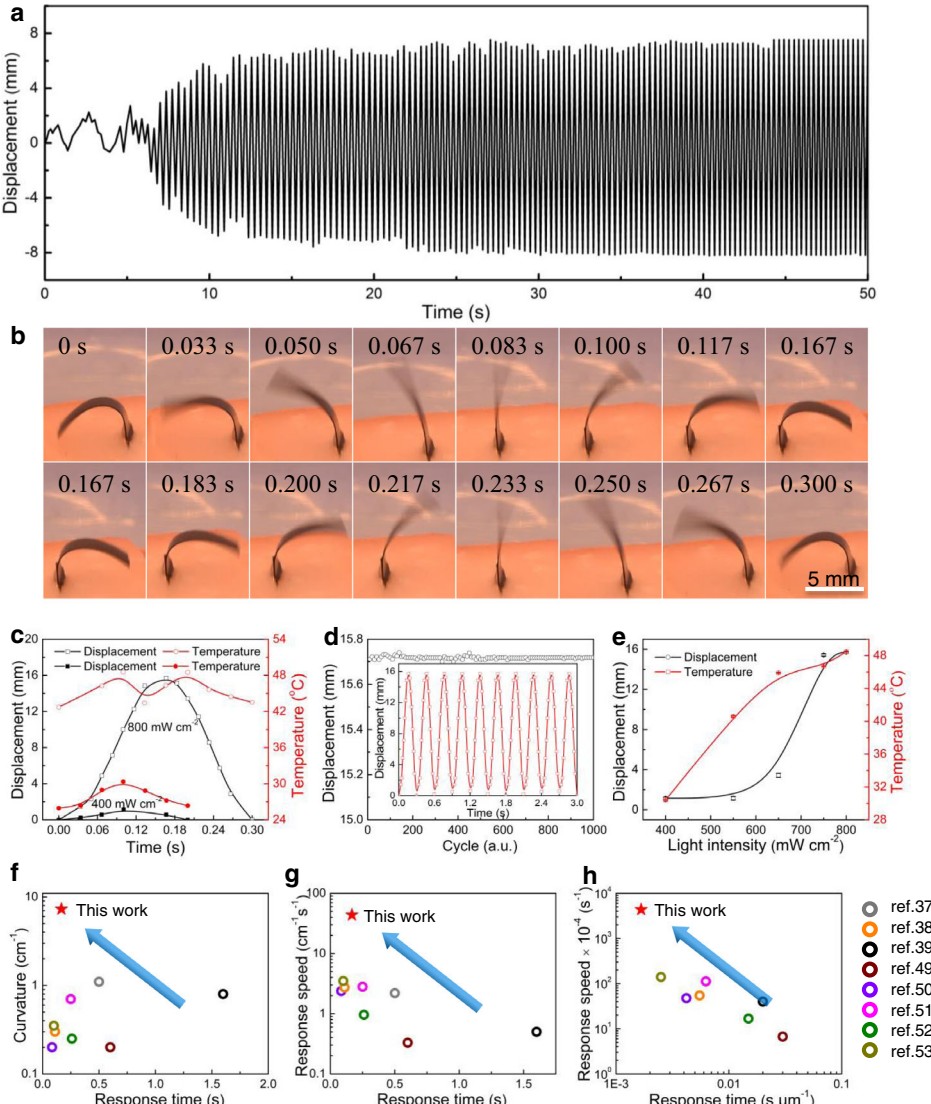

**Fig. 3 | Actuation performances of the PP/CB actuator under NIR light.**
**a** Displacement as a function of time during the oscillating actuation for the porous PP/CB film (9 mm × 3 mm × 100 μm) with continuous ethanol supply under 800 mW cm⁻² infrared light. **b** Photographs of the oscillating actuation under 800 mW cm⁻² infrared light. **c** Displacement and temperature as a function of time at 800 and 400 mW cm⁻² light intensities. **d** Cyclic tests for oscillating actuation. **e** Displacement, and the maximum temperature as a function of light intensity. **f**–**h** Comparison of the curvature and response time, response speed and their values normalized by the thickness with those of oscillating actuators[37–39,49–53]. Error bars denote the standard deviation.

× 100 μm) at different times with a light intensity of 800 mW cm⁻² during stable oscillation. The actuator completed one oscillation cycle in 0.3 s, with maximum bending curvature of 7.3 cm⁻¹ and displacement of 15.7 mm, with the vertical line as the oscillation center, and exhibited 0.9% light-to-work energy conversion efficiency (Supplementary Note 1), which was superior to that of reported light-responsive actuators, including polycarbonate/carbon nanotube (CNT) (0.01%)[60], CNT/polydimethylsiloxane (PDMS) (0.58%)[39], and Fe₃O₄ NPs@graphene oxide (GO) (0.38%)[61]. The corresponding infrared images during one oscillation cycle showed that the minimum temperature during actuation (42.7 °C) occurred at almost the maximum bending curvature (Supplementary Fig. 22), and the film showed the maximum temperature when it oscillated to the vertical position (48.5 °C) (Fig. 3c). This correlated well with the distance of the actuator to the light source, where a higher light intensity was obtained for a smaller distance to the light source (Supplementary Fig. 23). Stable oscillation was observed without decay of the actuation displacement during 1000 cycles of actuation with continuous ethanol supply, as shown in Fig. 3d. For a 0.1 mg oscillating actuator (9 mm × 3 mm ×

100 μm), 1.65 g of ethanol are used in the film and 7 g of ethanol are used in the reservoir. Because the actuator is in a sealed vessel, the evaporated ethanol can be recovered and flow back into the reservoir. Therefore, in theory almost no solvent waste occurs during such light-irradiated oscillating actuation. We investigated the influence of the film thickness of the actuator (60, 100, and 150 μm) on the actuation stress, amplitude, and frequency. The results showed that the actuator with film thickness of 60, 100, and 150 μm exhibited the actuation stress of 26.83, 26.72, and 26.67 kPa, the amplitude of 9.98, 15.73, and 14.15 mm, and the frequency of 3.3, 3.3, and 3.3 Hz, respectively. The 100 μm thick PP/CB film almost has the best oscillation performance among the films with different thicknesses (Supplementary Fig. 24). Therefore the 100 μm-thick actuator was employed. The film pore size is also an important factor affecting the actuation performance, which would be investigated and optimized in our future work in detail.

Decreasing the light intensity to 400 mW cm⁻² decreased the maximum bending displacement to 1.1 mm, and one oscillation cycle could be completed in 0.2 s (Fig. 3c). Different from the case of 800 mW cm⁻² infrared light, as the actuator film bent off the vertical

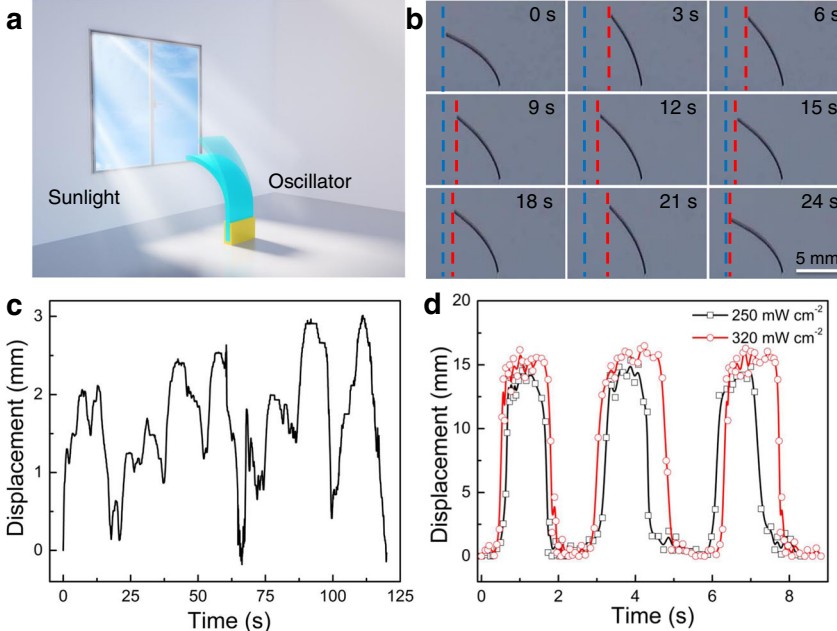

**Fig. 4 | Actuation performances of the PP/CB actuator under sunlight and simulated sunlight. a** Schematic illustration of the oscillating bending actuation of the porous film exposed to sunlight. **b**, **c** Photographs and displacement as a function of time for the oscillating actuation under approximately 50 mW cm⁻² sunlight. **d** Displacement as a function of time for the oscillating actuation under 320 and 250 mW cm⁻² sunlight simulated by a Xe lamp.

line under 400 mW cm$^{-2}$ infrared light, it only oscillated to this side and was not able to bend back to the other side (Supplementary Fig. 25). During oscillation, a much smaller temperature variation (between 25.9 and 30.3 °C) was observed between the maximum and minimum curvatures of the actuator (3.7 and 1.2 cm$^{-1}$, respectively). The environment condition is rather important affecting the actuation performance. For example, the oscillation actuation performance is influenced by the environment temperature. With the environmental temperature increased from 23 to 29 °C, the actuation amplitude of the PP/CB porous film increased from 0.62 to 13.46 mm under 400 mW cm$^{-2}$ NIR light irradiation, and the actuation frequency increased from 1.54 to 4.65 Hz with the environmental temperature increased from 23 to 25 °C, and then decreased to 1.21 Hz with the environmental temperature further increased to 29 °C (Supplementary Fig. 26). Increasing the light intensity from 400 to 650 mW cm$^{-2}$ dramatically increased the maximum temperature from about 30.3 to 46.0 °C and increased the temperature variation during actuation from 4.4 to 9.4 °C (Fig. 3e and Supplementary Fig. 27). This only slightly increased the bending displacement from 1.1 to 3.4 mm, and the actuator oscillated to one side of the vertical line. Interestingly, further increasing the light intensity to 750 mW cm$^{-2}$ resulted in oscillating actuation between the two sides of the vertical line, and the displacement largely increased to 15.4 mm, with the maximum temperature increasing to 46.6 °C, although the temperature variation during actuation dramatically decreased, with a minimum temperature of 40.6 °C (Supplementary Fig. 27). The maximum oscillation frequency (6 Hz) was obtained at 650 mW cm$^{-2}$, which showed the maximum temperature variation (9.4 °C) among the different light intensities (Supplementary Fig. 28). The obtained experimental frequency was between 3.3 and 6 Hz, which is in good agreement with the theoretical natural resonance frequency (Supplementary Note 2). Thermally operated actuator usually suffers from the slower response time because the cooling takes quite bit of time while the initial heating can be done very fast. For the actuation performance optimization, the simultaneous fast heating and cooling as well is very important. In this work, the cooling speed of the PP/CB oscillator actuator can be high benefiting from the solvent-evaporation aided heat dissipation, and

therefore this solvent-aided oscillating is suitable to obtain fast actuation frequency. The displacement and oscillating frequency are strongly affected by the film size (length and width), mechanical properties of the actuating film, solvent types, volume expansion coefficient, the wetting height of the solvent in the film, and light intensity (Supplementary Fig. 29). In addition, the oscillating capacity would be highly affected by the pore size and pore types of the film, film mass, wavelength of the incident light, light absorption and thermal conversion capacity of the film. To improve the actuation frequency, the following factors need to be considered, including increasing the capacity of solvent absorption and evaporation, increasing the elasticity of the actuation film, and the volume change capacity during actuation. Overall, the oscillation amplitude (bending curvature or bending angle) and actuation speed at a shorter response time, and their values normalized by the thickness are also superior to those of most oscillators (Fig. 3f–h, Supplementary Fig. 30, and Table 1)[37–39,49–53]. Although the actuation speed and bending angle are among the best of the bimorph actuators, the energy conversion efficiency (0.9%) is comparable or in the same level to paraffin wax-CNT hybrid yarn actuator (0.5%)[62], silicone-CNT hybrid yarn actuator (4.3%)[63], graphdiyne actuator (6.03%)[28], and sheath-run CNT artificial muscles actuator (3.8%)[64]. Because of the porous structure, the generated stress (26.72 kPa) of the PP/CB oscillating actuator are smaller than those of other types of bimorph actuators, e.g., Cu@polyvinyl alcohol-co-polyethylene/GO actuator (1.35 MPa)[65], liquid crystal network actuator (7 MPa)[66], poly(indenoquinacridone)/CNT actuator (3.2 MPa)[67], and CNT/rGO actuator (16 MPa)[68]. In addition, continuous supply of solvent is needed to obtain uninterrupted oscillatory actuation, to solve this problem, a design of a cycle cooling system in a sealed vessel would be an option for continuous supply of evaporation of solvent.

We then tried to understand the actuation mechanism based on the oscillation behavior at different light intensities. We first analyzed why the actuator only oscillated to one side of the vertical line at relatively low light intensities. A vertically positioned dry porous PP film was manually bent to different curvatures, and upon removing the bending force it fully recovered to the vertical position due to the

built-in internal stress ($F_1$) induced by deformation (Supplementary Fig. 31A). However, a vertically positioned porous PP/CB film infiltrated with ethanol was not able to recover to the vertical position upon removal of the bending force (Supplementary Fig. 31B). This indicated that the gravity ($G$) of the PP/CB film infiltrated with ethanol inhibited unbending of the porous PP film to the vertical position. Under continuous irradiation with infrared light, ethanol evaporation was sped up, resulting in an increased volume decrease of the actuator, causing an additional force ($F_2$) to help the actuator bend back to the vertical line. This additional force $F_2$ caused by photothermal-induced solvent evaporation would increase with increasing light intensity because a higher light intensity caused increased volume shrinkage of the actuator. Because $F_1$, $F_2$, and $G$ varied during unbending of the actuator, oscillating actuation between two critical curvatures was observed when the infrared light was on, and the actuator was not able to bend back to the other side until the force $F_2$ was large enough to reach a critical point (Fig. 1a).

We then investigated the general applicability of oscillating actuation for different solvents. Bending actuation occurred for the horizontally placed actuator film irradiated with vertical infrared light and infiltrated with polar solvents with low boiling points, such as ethanol, methanol, tetrahydrofuran (THF), acetone, ethyl acetate, and dichloromethane. The actuator with ethanol, acetone, and ethyl acetate showed a much larger bending curvature than that with dichloromethane, methanol, and THF (Supplementary Fig. 32). Interestingly, continuous supply of ethanol, methanol, and THF also showed faster oscillation with larger displacement than that of dichloromethane, acetone, and ethyl acetate (Supplementary Fig. 33). Polar solvents with high boiling points ($N,N$-dimethylformamide and dimethyl sulfoxide) and nonpolar solvents (petroleum ether, cyclohexane, and toluene) provided negligible bending actuation, possibly because the slow evaporation rate (for the high-boiling-point solvents) or the low wicking height and poor wetting (for the nonpolar solvents) resulted in a small volume change under light illumination. In addition, we fabricated a porous PDMS film, which could generate oscillation with ethanol and ethyl acetate supply (Supplementary Fig. 34).

## Oscillating actuation under sunlight and simulated sunlight

Oscillating actuation could occur indoors under sunlight irradiation through a window (Fig. 4a). A series of snapshots depict the irregular oscillating actuation of the porous film (Fig. 4b). Under ethanol supply, it oscillated to one side of the vertical line, with a maximum bending displacement of 3.0 mm, under sunlight exposure in a window transmission measurement with an approximately 50 mW cm$^{-2}$ light intensity and a temperature of approximately 30 °C (Fig. 4c). Although it has an irregular oscillation amplitude and a slow oscillation frequency, this is an oscillating actuator driven by solvent evaporation under solar light. The oscillating displacement enhanced to 9.6 mm when the experiment was directly performed under the intense sunlight (100 mW cm$^{-2}$) with a temperature of 36 °C, and no performance decay was observed for 20 days (Supplementary Fig. 35). We then tested the oscillating actuation under simulated solar light with different light intensities using a xenon (Xe) lamp. Periodic oscillatory vibration around the vertical line was obtained for light intensities of 250 and 320 mW cm$^{-2}$, with displacements of 15.1 and 16.2 mm, respectively, and oscillation periods of 2.13 and 2.06 s, respectively (Fig. 4d). Different from the case of infrared light irradiation, the actuator film stayed at the vertical position and at the maximum bending curvature for a while, where secondary oscillation with much a smaller displacement and a shorter period was observed. This multilevel oscillating actuation should be ascribed to the parallel incident light generated by the Xe lamp. Decreasing the light intensity (e.g., to 130 and 200 mW cm$^{-2}$) caused oscillation to one side of the vertical line, where the film temperature at a 200 mW cm$^{-2}$ light intensity during actuation was much smaller than for a light intensity of 320 mW cm$^{-2}$

(Supplementary Fig. 36). Besides, we added a performance test experiment using incident light with different band wavelengths (458, 465, 513, 589, and 632 nm), which exhibited actuation amplitudes of 0.5, 0.25, 0.35, 0.24, and 0.4 mm, actuation frequencies of 8.75, 7.83, 6.75, 7.83, and 7 Hz, and actuation stresses of 19.38, 19.42, 19.57, 18.5, and 18.15 kPa (Supplementary Fig. 37).

## Work capacity of the light engine

In this section, we investigated whether the oscillating actuator can be used to output work by carrying a load. As expected, fast oscillation occurred when the actuator film carried a 7.5 mg piece of foam or a 7.1 mg film under 800 mW cm$^{-2}$ infrared light illumination, corresponding to weight-lift ratios of 2 and 1.9, respectively (Fig. 5a, Supplementary Fig. 38 and Movie 2). Increasing the weight-lift-ratio from 0.5 to 4 slightly decreased the displacement from 17.9 to 13.3 mm, and the oscillation period varied from 0.56 to 0.95 s. Further increasing the weight-lift ratio to 6 resulted in oscillation to one side of the vertical line, and the displacement decreased to 2.5 mm (Fig. 5b). We then calculated how much work could be generated during oscillation of the actuator with a hanging load. The specific work of the actuator ($W_{act}$) and the specific power ($P_{act}$) were calculated using the following equations: $W_{act} = m \cdot \mathbf{g} \cdot h / m_0$ and $P_{act} = m \cdot \mathbf{g} \cdot h / (m_0 \cdot t)$ ($m$ and $m_0$ are the masses of the load and actuator, respectively; $h$ is the height of the load; and $t$ is the time to lift the load to the maximum height). The specific work increased from $3.5 \times 10^{-5}$ J g$^{-1}$ to a maximum value of $12.0 \times 10^{-5}$ J g$^{-1}$ as the weight-lift ratio increased from 0.5 to 4, and then decreased to 0 J g$^{-1}$ at the weight-lift ratio of 6. The maximum specific power ($2.0 \times 10^{-4}$ W g$^{-1}$) was also obtained at a weight-lift ratio of 4 (Fig. 5c).

Trains use an internal combustion engine to generate rotary motion of the wheels by burning gasoline. Here, we designed a similar strategy to generate mechanical work by using oscillating bending actuation, where a load was tethered to the actuator film through connection with a thread (Fig. 5d). The oscillation of the actuator could cyclically lift up the load and output work. The mechanical work generated during actuation could be calculated by the change in the gravitational potential energy of the load when the load was lifted up during bending of the actuator from one side to the other side, and the power was calculated as the mechanical work divided by the time for a half cycle of the actuation. The maximum specific work ($30.9 \times 10^{-5}$ J g$^{-1}$) and the maximum specific power ($15.4 \times 10^{-5}$ W g$^{-1}$) were obtained for a load mass of 1.9 mg (Fig. 5e), where the displacement and the oscillation period of the actuator were 12.4 mm and 1.28 s, respectively. Such a load could be replaced with an eccentric shaft so that oscillating bending could be transferred to the rotation of a wheel. The output work and the generated mechanical power due to the light-induced solvent evaporation of a film are currently investigated. Future work would be undertaken to further increase the actuation speed, mechanical work and power density, and energy efficiency.

## Discussion

Here we create a solar engine by using an oscillating actuator, which shows large oscillation displacement and high bending amplitude compared to the previous bimorph actuators. This solvent-assisted light-driven oscillator is realized by a porous film, and can respond to light irradiation from different angles.

The porous structure of the film allows fast absorption/desorption of organic solvents, and photothermal irradiation speeds up solvent evaporation and results in asymmetric film volume expansion. These combined effects lead to oscillating motions. Consequently, obvious self-oscillation behavior occurs even under the stray light in a house.

The actuator can even oscillate while carrying a load or lifting a load under light irradiation. This provides a way to output mechanical work by directly harvesting solar energy by employing bimorph

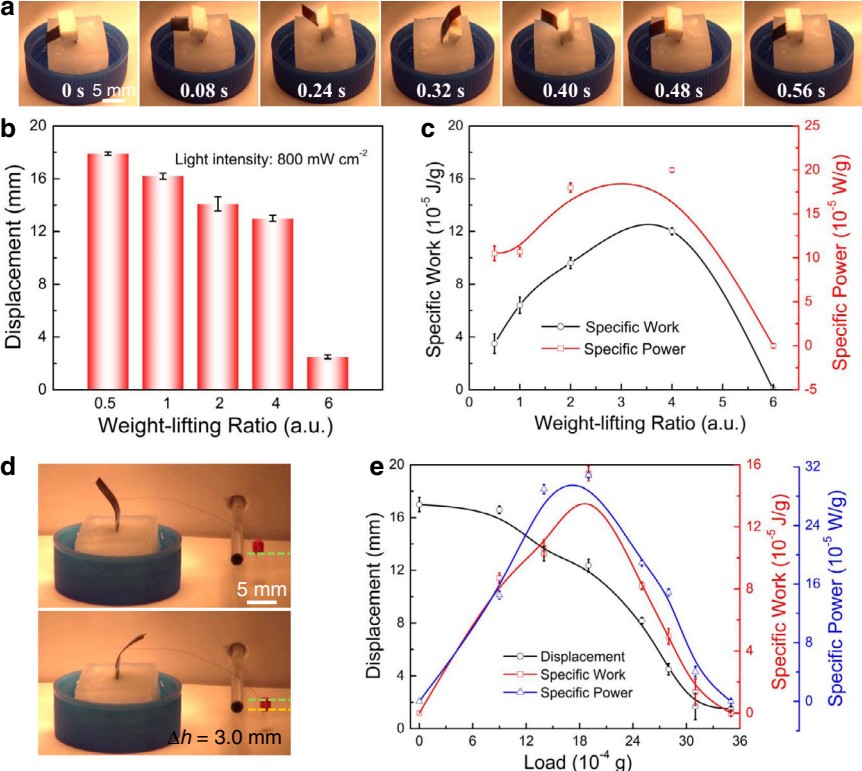

**Fig. 5 | Work capacity of the light engine. a** Photographs of oscillating film loaded with an object. **b** Displacement with different weight-lift ratios. **c** Work and power as a function of the weight-lift ratio upon infrared irradiation. **d** Photographs of film as a light motor. **e** Displacement, work, and power as a function of load. Error bars denote the standard deviation.

actuators. This provides the possibility of aircrafts, vehicles, and soft robotics working without the supply of fossil energy. For example, such a solar engine could be used in universe exploration, mars or moon landings, and searching, probing, or rescue in difficult-to-reach areas for human beings.

The design would inspire a wide variety of branches in machinery design, biomedicine, sensors and detectors, wearable devices, and miniaturized robotics. The synergistic design of an asymmetric porous structural change assisted by solvent wetting would produce possibilities and properties when adopted for different types of functional materials. For example, a solar energy soft robot would become possible by combining actuation, sensing, and solar energy harvesting.

The current research would inspire inter-disciplinary scientific application scenarios would be developed, for example, in the fields of biomedical, flexible photovoltaic devices, electricity generation, production line with sustainable energy, and heat conduction, magnetic and acoustic applications, etc. In addition, the study on the multi-disciplinary project would produce fundamental discoveries on the mechanical, interfacial, and chemical properties.

In summary, self-oscillating actuators were achieved under divergent light, including sunlight, simulated sunlight, and infrared light, by using photothermal- induced alternating solvent evaporation from a porous PP/CB film. The actuator showed the largest oscillation displacement (15.7 mm) and amplitude (7.3 cm$^{-1}$ or 224°) at a smaller temperature change (5.8 °C) compared to the oscillating actuators driven by light reported thus far. The excellent self-oscillation performance originated from the indispensable combination of the fast volume shrinkage and recovery due to the fast solvent evaporation and supply, appropriate modulus, and high flexibility of the PP/CB film, and the oscillation can never stop under continuous solvent supply. In addition, oscillating actuation can occur for the PP/CB film at different tilt angles under divergent light with different incident angles, and is applicable to a variety of volatile polar solvents, which can be used for leakage monitoring of volatile organic vapor. Moreover, oscillating actuation can still occur when carrying a load and can be used in different scenarios such as signal transmission, sensors, and controllers by replacing the loads with different functional components. The oscillating actuation can open or close a valve under light irradiation, and therefore serves as a controller to control the transmission of liquid, gas, or even light. Because the oscillating amplitude and frequency is sensitive to the light source and different types of organic solvents, it can serve as a sensor to the light intensity and wavelength by employing different types of pigments. The oscillating actuation during light transmission can be used to modulate the light signal during signal transmission.

## Methods

### Fabrication of the porous film oscillating actuator

Commercial porous PP/CB films were used to show the general applicability of this oscillating actuation realized by photothermal solvent evaporation, which included PP/CB film (Nitto, SCF serials, 60, 100, and 150 μm). In addition, a PDMS porous film was prepared by adding sodium chloride (NaCl, 0.06 g) and CB (0.0075 g) powder into a PDMS mixture (0.33 g) composed of the base agent and curing agent (10:1) and cured to form a dense film, followed by dissolving NaCl to generate pores. For oscillating actuation, porous films with the desired size were supplied with different solvents, including ethanol, methanol, ethyl acetate, dichloromethane, THF, and acetone. For comparison, photothermal actuation of the porous films infiltrated with solvents without a continuous solvent supply was also characterized.

### Characterizations

The weight loss by solvent evaporation was measured on a balance with or without light irradiation. The contact angle was obtained on contact angle goniometer (Zhongchen, JC2000D1). The SEM images were obtained on a scanning electron microscopy (MERLIN Compact).

The swelling ratios were obtained by measuring the size of the film before and after immersion in solvents on a metallographic microscope (Chenxing, CXML1000). The UV–Vis absorption spectrum was obtained on a spectrometer (PerkinEelmer, Lambda 950). The FTIR spectra were obtained on a Fourier transform infrared spectrometer (Thermo fisher Scientific, Nicolet iS50). The wicking height of the solvent on the film was obtained by a video camera. The pore size distribution was obtained on a mercury intrusion meter (PoreMaster, GT60) and on a BET (Gemini 2390). The stress–strain curves were obtained on a mechanical tester (Moxin, MX-0508). The thermal images and the temperature of the actuator were obtained by the infrared thermal imager (FLIR, T440). The simulated sunlight was obtained by a Xe lamp (Zhongjiaojinyuan, CEL-HXF300).

## Data availability

The authors declare that all data supporting the findings of this study are available within the main text and Supplementary Information. Source data are provided with this paper.

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

## Acknowledgements

This work was supported by the National Key Research and Development Program of China (grants 2019YFE0119600), the National Natural Science Foundation of China (grants 52090034, 51973093, and 51773094), Frontiers Science Center for New Organic Matter, Nankai University (Grant Number 63181206), the National Special Support Plan for High-Level Talents People (grant no. C041800902), the Science Foundation for Distinguished Young Scholars of Tianjin (grant no. 18JCJQJC46600), the Fundamental Research Funds for the Central Universities (grant 63171219), the State Key Laboratory for Modification of Chemical Fibers and Polymer Materials, Donghua University (grant LK1704), the Key Scientific Research Projects of Colleges and Universities in Henan Province (grant number 21A150006), and the Industrial Research Project of Technology Bureau of Anyang.

## Author contributions

Z.F.L. and J.J.L. were responsible for the experimental concept and design. J.J.L. and L.L.M. carried out the most experiments, characterization and data analyses. Z.F.L., X.Z., and Y.S.C. was responsible for project administration, conceptualization, supervision, formal analysis, funding acquisition, validation, writing original draft, review and editing. All authors wrote the paper. All authors provided comments and agreed with the final form of the manuscript.

## Competing interests

The authors declare no competing interests.
