## [Peer Review File · Nature Communications]

Oscillating light engine realized by photothermal solvent evaporationREVIEWER COMMENTS

Reviewer #1 (Remarks to the Author):

In this manuscript, the authors propose a unique concept of oscillating bending polymeric actuator which is driven by near-infrared light-thermal conversion. The authors found that when the light is irradiated from above to a standing porous film with the condition that one side of the film is immersed in a solvent, continuous self-vibration is caused.

The principle of self-vibration mechanism is simple and understandable; the swollen porous polymer films, shrink by heating using photothermal conversion of incorporating carbon materials with near-infrared Near-infrared light-irradiation. When shrinkage due to solvent evaporation occurs on one side of the film, light hits the outer surface that did not shrink, causing reverse shrinkage. Continuous separately excited oscillation is realized by constantly supplying the solvent to compensate for consumption.

As the authors claim, they have proved to be able to be achieved stable high-speed vibration that has never been achieved as a photo-excited actuator. Furthermore, they provided two types of porous polymer films based on PDMS or ethyl polyacrylate, demonstrating the universality of the concept.

There seems to be no criticism in terms of the experimental and its interpretation. I recommend acceptance subject to minor revision, as described hereafter.

1) This new type of actuator is also considered to have drawbacks, such as the need for a continuous supply of solvent, the expected low energy efficiency, and the smallness of generated stress. The authors should discuss those in comparison with other actuators.

2) In conclusion, the authors have listed many applicational directions via the micro transport of objects.

They should explain briefly how the vibration leads to micro-transportation. In addition, it is difficult to imagine how a simple vibration mechanism can be applied to the following applications, such as signal transmission, sensors, and controllers. A certain explanation is needed.

3) There seems to be a bias in the reference papers cited. They should refer to the literature related to photothermal conversion actuators widely.

For instance,

@ Near-infrared response by mixing carbon with PDMS for the first time:

S. V. Ahir et al., Nature Mater. 2005, doi: 10.1038 / nmat1391

@ Example of separately excited oscillation using photothermal conversion on the water surface:

Y. Harada et al., J. Phys. Chem. C 2018, DOI: 10.1021 / acs.jpcc.7b11123

@ An example of research that brings out different local movements with unfocused light:

S. Watanabe et al., Sci. Rep. 2018, DOI: 10.1038 / s41598-018-31932-2

@ An example of soft actuator using a liquid-to-gas phase change

S. Eristoff et al., Adv. Mater 2022, DOI: 10.1002 / adma.202109617

Reviewer #2 (Remarks to the Author):

This study presents an interesting study on a solar engine using an oscillating actuator, which was realized through an alternating volume decrease of each side of a polypropylene/carbon black polymer film induced by photothermal-derived solvent evaporation. The phenomena look unique and the application in various soft robotics is expected to be very promising.

This can be recommended for publication with the following revision comments,

1) Inspired by the natural processes, energy conversion from solar to electric and thermal energy is definitely an interesting way to develop a new actuator. I guess the current process use a photo-thermal process. In the various living organisms using solar energy, some of them use photo-chemical process. This also can be briefly discussed in the introduction part.

2) A general strategy is to confine the incident light in a localized region so that the light-induced bending causes the actuator to be no longer illuminated by the light beam and bend back to be

illuminated again. This requires the incident light to have a very strict beam width and a special irradiation angle. As sunlight does not have a strict beam width and the incident angle varies with time, the actuator has difficulty escaping from sunlight irradiation during actuation. Here, this characteristics (no angle dependence) of the sun light is called 'diffuse'.

3) The decreased volume should quickly recover when there is no light illumination. The time for volume recovery and shrinkage would match so that the volume can be fully recovered before the next actuation cycle. What is the most important factor limiting the actuation frequency? And how this can be enhanced?

4) To obtain efficient conversion from solar to mechanical energy, the oscillating actuation performance (e.g., bending amplitude, frequency, and output work) needs to be optimized. This requires delicate design of a synergistic combination of actuator mechanical properties (e.g., modules, elasticity, and size), light absorption and conversion, and actuation stress. Thermally operated actuator usually suffers from the slower response time because the cooling takes quite bit of time while the initial heating can be done very fast. For the actuation performance optimization, the simultaneous fast heating and cooling as well is very important. This also needs to be discussed.

5) The major application area of thin film actuator is either actuation or morphing. Actuator usually needs large force to do a certain function while the morphing (Materials Today, 41, 243 (2020)) usually needs smaller force just to deform itself. Because morphing application usually needs smaller force and energy, it has a relatively thinner film thickness compared with actuation application. This needs to be briefly discussed in the introduction part.

6) When a vertically positioned porous film with continuous solvent supply was irradiated with light, the initial deflection of the film from the central point caused nonequal solvent evaporation from the two sides and asymmetric volume shrinkage. This phenomenon resulted in the actuator bending towards the light and exposing the other side of the film to the light. The evaporation will be largely affected by the environment. How was the influence from the environment condition change to the actuator performance?

7) 100 micron-thick film investigated here showed a hierarchical porous structure with microscale average pore size of 40 micron, measured by the mercury intrusion method, and a nanoscale average pore size of approximately 91.83 nm. Usually, thinner actuator may show faster actuation. However, the final blocking force will be decreased. I wonder how the specific geometries (thickness, pore size etc) were decided.

8) The photothermal actuation can be done with the monochromic (or band wavelength) light. I wonder whether the wide band or narrow band is better for efficient actuation.

9) The prolonged use under the intense sunlight may induce the degradation (chemical or thermal) of the polymer thin film actuator. Any long term actuation under the intense light?

10) Scale bars are missing in pictures in figure 3B, 4B, 5A,D.

Reviewer #3 (Remarks to the Author):

This submission reports the preparation of photo-responsive bending actuators comprising the polymer thin film and CB. The volume of the film could be controlled by absorbing/evaporating the solvent and this process occurred reversibly induced by the generated heat from photothermal conversion. Finally, the fabricated actuators exhibited self-oscillating behavior that is achieved under light irradiation. Whereas the technical points are well supported by the data and analysis, the originality and significance of the study are not well explained to fulfill the requirement of a high standard of the journal.

Some specific points

1. There are more papers regarding self-oscillating photo-actuators that should be cited.
 - Y. Hu, Q. Ji, M. Huang, L. Chang, C. Zhang, G. Wu, B. Zi, N. Bao, W. Chen, Y. Wu, *Angew. Chem. Int. Ed.* 2021, 60, 20511.
 - Hu, Z., Li, Y. & Lv, Ja. Phototunable self-oscillating system driven by a self-winding fiber actuator. *Nat Commun* 12, 3211 (2021).
2. The originality of the paper is not well described. Light-driven, photo-thermal, solvent-

evaporation induced, self-oscillating actuators are well known in this field. It is hard to find any scientific finding to justify the originality of the submission.

3. The authors mentioned that the fabricated photo-actuators could be operated under sunlight, however, the displacement of the actuators are too modest and slow, as shown in Figure 4b and c.

4. It is not provided how much volume of the solvent is needed to operate and self-oscillate the actuators in Figure 3a. It seems that there is much amount of solvent waste even though the actuators are just waiting for the operation.

5. Fig 3a. The bending oscillation of the actuators is stable and reliable over 10s. Moreover, the actuation of the first tens of cycles was not successful. Why? Is it always observed for every new session?

6. What is the role of some additives (Ca, Ti)?

Response to the reviewers:

Response to the reviewer #1:

In this manuscript, the authors propose a unique concept of oscillating bending polymeric actuator which is driven by near-infrared light-thermal conversion. The authors found that when the light is irradiated from above to a standing porous film with the condition that one side of the film is immersed in a solvent, continuous self-vibration is caused.

The principle of self-vibration mechanism is simple and understandable; the swollen porous polymer films, shrink by heating using photothermal conversion of incorporating carbon materials with near-infrared light-irradiation. When shrinkage due to solvent evaporation occurs on one side of the film, light hits the outer surface that did not shrink, causing reverse shrinkage. Continuous separately excited oscillation is realized by constantly supplying the solvent to compensate for consumption.

As the authors claim, they have proved to be able to be achieved stable high-speed vibration that has never been achieved as a photo-excited actuator. Furthermore, they provided two types of porous polymer films based on PDMS or ethyl polyacrylate, demonstrating the universality of the concept.

There seems to be no criticism in terms of the experimental and its interpretation. I recommend acceptance subject to minor revision, as described hereafter.

Response: We thank the reviewer for these insightful comments, and we have made a substantial revision to improve this manuscript based on the reviewers' comments.

1) This new type of actuator is also considered to have drawbacks, such as the need for a continuous supply of solvent, the expected low energy efficiency, and the smallness of generated stress. The authors should discuss those in comparison with other actuators.

Response: We thank the reviewer for this suggestion. We compared the characteristics of the PP/CB actuator with other actuators, and added the discussion in the revised manuscript, as follows (Page 12, Line 339-350):

Although the actuation speed and bending angle are among the best of the bimorph actuators, the energy conversion efficiency (0.9%) is comparable or in the same level to paraffin wax-CNT hybrid yarn actuator (0.5%)⁶², silicone-CNT hybrid yarn actuator (4.3%)⁶³, graphdiyne actuator (6.03%)²⁸, and sheath-run CNT artificial muscles actuator (3.8%)⁶⁴. Because of the porous structure, the generated stress (26.72 KPa) of the PP/CB oscillating actuator are smaller than those of other types of bimorph actuators, e.g., Cu@polyvinyl alcohol-co-polyethylene/GO actuator (1.35 MPa)⁶⁵, liquid crystal network actuator (7 MPa)⁶⁶, poly(indenoquinacridone)/CNT actuator (3.2 MPa)⁶⁷, and CNT/rGO actuator (16 MPa)⁶⁸. In addition, continuous supply of solvent is needed to obtain uninterrupted oscillatory actuation, to solve this problem, a design of a cycle cooling system in a sealed vessel would be an option for continuous supply of evaporation of solvent.

References:

62. Lima, M. D. et al. Electrically, chemically, and photonically powered torsional and tensile actuation of hybrid carbon nanotube yarn muscles. *Science* **338**, 928–932 (2012).
63. Lima, M. D. et al. Efficient, absorption-powered artificial muscles based on carbon nanotube hybrid yarns. *Small* **11**, 3113–3118 (2015).
64. Mu, J. K. et al. Sheath-run artificial muscles. *Science* **365**, 150–155 (2019).
65. Wang, W. et al. Graphene oxide/nanofiber-based actuation films with moisture and photothermal stimulation response for remote intelligent control applications. *ACS Appl. Mater. Interfaces* **13**, 48179–48188 (2021).
66. Lu, X. L. et al. Tunable photocontrolled motions using stored strain energy in malleable azobenzene liquid crystalline polymer actuators. *Adv. Mater.* **29**, 1606467 (2017).
67. Yu, K. Q. et al. Robust jumping actuator with a shrimp-shell architecture. *Adv. Mater.* **33**, 210455 (2021).
68. Qiao, J. et al. Large-stroke electrochemical carbon nanotube/graphene hybrid yarn muscles. *Small* **14**, 180188 (2018).

2) *In conclusion, the authors have listed many applicational directions via the micro*

transport of objects. They should explain briefly how the vibration leads to micro-transportation. In addition, it is difficult to imagine how a simple vibration mechanism can be applied to the following applications, such as signal transmission, sensors, and controllers. A certain explanation is needed.

Response: We thank the reviewer for this kind suggestion. We re-write the discussion as follows.

Moreover, oscillating actuation can still occur when carrying a load and can be used in different scenarios such as signal transmission, sensors, and controllers by replacing the loads with different functional components. The oscillating actuation can open or close a valve under light irradiation, and therefore serves as a controller to control the transmission of liquid, gas, or even light. Because the oscillating amplitude and frequency is sensitive to the light source and different types of organic solvents, it can serve as a sensor to the light intensity and wavelength by employing different types of pigments. The oscillating actuation during light transmission can be used to modulate the light signal during signal transmission. We added this discussion in Page 17, Line 491-499.

3) There seems to be a bias in the reference papers cited. They should refer to the literature related to photothermal conversion actuators widely.

For instance,

@ Near-infrared response by mixing carbon with PDMS for the first time:

S. V. Ahir et al., Nature Mater. 2005, doi: 10.1038/nmat1391

@ Example of separately excited oscillation using photothermal conversion on the water surface:

Y. Harada et al., J. Phys. Chem. C 2018, DOI: 10.1021/acs.jpcc.7b11123

@ An example of research that brings out different local movements with unfocused light:

S. Watanabe et al., Sci. Rep. 2018, DOI: 10.1038/s41598-018-31932-2

@ An example of soft actuator using a liquid-to-gas phase change

S. Eristoff et al., Adv. Mater 2022, DOI: 10.1002/adma.202109617

Response: We added these important relevant literatures for the photothermal and electrothermal actuators, as follows (Page 2, Line 58-61)

To realize continuous work output of an actuator, a pulsed stimulus is generally required by periodically switching on/off the supply of moisture^{26,27}, electricity^{28,29}, light³⁰⁻³⁴, etc., or by using an alternating current electrical supply.

References:

29. Amjadi, M. & Sitti, M. Self-sensing paper actuators based on graphite-carbon nanotube hybrid films. *Adv. Sci.* **5**, 1800239 (2018).
31. Ahir, S. V. et al. Photomechanical actuation in polymer-nanotube composites. *Nat. Mater.* **4**, 491–495 (2005).
32. Harada, Y. et al. Emergence of pendular and rotary motions of a centimeter-sized metallic sheet under stationary photoirradiation. *J. Phys. Chem. C* **122**, 2747–2752 (2018).
33. Watanabe, S., Era, H. & Kunitake, M. Two-wavelength infrared responsive hydrogel actuators containing rare-earth photothermal conversion particles. *Sci. Rep.* **8**, 13528 (2018).
34. Eristoff, S. et al. Soft actuators made of discrete grains. *Adv. Mater.* **34**, 2109617 (2022).

Response to the reviewer #2:

This study presents an interesting study on a solar engine using an oscillating actuator, which was realized through an alternating volume decrease of each side of a polypropylene/carbon black polymer film induced by photothermal-derived solvent evaporation. The phenomena look unique and the application in various soft robotics is expected to be very promising.

This can be recommended for publication with the following revision comments,

Response: We thank the reviewer for these valuable comments and insightful suggestions, and we have carefully addressed these questions in the revised version of this article. Here we provide the point-to-point reply below.

1) Inspired by the natural processes, energy conversion from solar to electric and thermal energy is definitely an interesting way to develop a new actuator. I guess the current process use a photo-thermal process. In the various living organisms using solar energy, some of them use photo-chemical process. This also can be briefly discussed in the introduction part.

Response: Thank you for the suggestion. We added the discussion for the solar energy to chemical energy conversion (as ref. 6-8) in the introduction part. (Page 2, Line 40)

References:

6. Qiu, B. C. et al. Integration of redox cocatalysts for artificial photosynthesis. *Energy Environ. Sci.* **14**, 5260–5288 (2021).

7. Cai, T. et al. Cell-free chemoenzymatic starch synthesis from carbon dioxide. *Science* **375**, 1523–1527 (2021).

8. Liu, J. T. et al. Polychromatic solar energy conversion in pigmentprotein chimeras that unite the two kingdoms of (bacterio)chlorophyll-based photosynthesis. *Nat. Commun.* **11**, 1542 (2020).

2) A general strategy is to confine the incident light in a localized region so that the light-induced bending causes the actuator to be no longer illuminated by the light beam and bend back to be illuminated again. This requires the incident light to have a very

strict beam width and a special irradiation angle. As sunlight does not have a strict beam width and the incident angle varies with time, the actuator has difficulty escaping from sunlight irradiation during actuation. Here, this characteristics (no angle dependence) of the sun light is called 'diffuse'.

Response: We thank the reviewer for this good suggestion. We added the statement on “diffuse” characteristics of sunlight in the revised manuscript, as follows: (Page 3, Line 68-70)

As sunlight is characterized as diffuse light, which does not have a strict beam width and the incident angle varies with time, the actuator has difficulty escaping from sunlight irradiation during actuation.

3) The decreased volume should quickly recover when there is no light illumination. The time for volume recovery and shrinkage would match so that the volume can be fully recovered before the next actuation cycle. What is the most important factor limiting the actuation frequency? And how this can be enhanced?

Response: This is a good suggestion. The displacement and oscillating frequency are strongly affected by the film size (length and width), mechanical properties of the actuation film, solvent types, volume expansion coefficient, the wetting height of the solvent in the film, and light intensity (Supplementary Fig. 29). In addition, the oscillating capacity would be highly affected by the pore size and pore types of the film, film mass, wavelength of the incident light, light absorption and thermal conversion capacity of the film. To improve the actuation frequency, the following factors need to be considered, including increasing the capacity of solvent absorption and evaporation, increasing the elasticity of the actuation film, and the volume change capacity during actuation. We added the discussion in the revised manuscript. (Page 12, Line 326-335)

Figure S29. Displacement and oscillation frequency as a function of actuating film with different lengths (A), widths (B), wetting heights of the solvent in the film (C), light intensities (D). The original size of PP/CB film is 9 mm × 3 mm × 100 μm.

4) To obtain efficient conversion from solar to mechanical energy, the oscillating actuation performance (e.g., bending amplitude, frequency, and output work) needs to be optimized. This requires delicate design of a synergistic combination of actuator mechanical properties (e.g., modulus, elasticity, and size), light absorption and conversion, and actuation stress. Thermally operated actuator usually suffers from the slower response time because the cooling takes quite bit of time while the initial heating can be done very fast. For the actuation performance optimization, the simultaneous fast heating and cooling as well is very important. This also needs to be discussed.

Response: Thank you for this good suggestion. We added this important point in both the introduction part as well as in the discussion part, as follows:

“This requires delicate design of a synergistic combination of actuator mechanical properties (e.g., modulus, elasticity, and size), light absorption and conversion, heating

and cooling rates, and actuation stress.” (Page 3, Line 81-84)

“Thermally operated actuator usually suffers from the slower response time because the cooling takes quite bit of time while the initial heating can be done very fast. For the actuation performance optimization, the simultaneous fast heating and cooling as well is very important. In this work, the cooling speed of the PP/CB oscillator actuator can be high benefiting from the solvent-evaporation aided heat dissipation, and therefore this solvent-aided oscillating is suitable to obtain fast actuation frequency.” (Page 11, Line 320-326)

5) The major application area of thin film actuator is either actuation or morphing. Actuator usually needs large force to do a certain function while the morphing (Materials Today, 41, 243 (2020)) usually needs smaller force just to deform itself. Because morphing application usually needs smaller force and energy, it has a relatively thinner film thickness compared with actuation application. This needs to be briefly discussed in the introduction part.

Response: We thank the reviewer for this valuable comment. We added this discussion in the context and cited the relevant literature for these statements, as follows (Page 2, Line 52-58)

The major application area of thin film actuator is either actuation or morphing. Actuator usually needs large force to do a certain function while the morphing usually needs smaller force just to deform itself²⁵. Because morphing application usually needs smaller force and energy, it has a relatively thinner film thickness compared with actuation application. In this work, in order to obtain output work, a high actuation stress, a large morphing amplitude, as well as a high actuation frequency, are highly desired.

Reference:

25. Kim, H. et al. Shape morphing smart 3D actuator materials for micro soft robot. *Mater. Today* **41**, 243-269 (2020).

6) When a vertically positioned porous film with continuous solvent supply was irradiated with light, the initial deflection of the film from the central point caused nonequal solvent evaporation from the two sides and asymmetric volume shrinkage. This phenomenon resulted in the actuator bending towards the light and exposing the other side of the film to the light. The evaporation will be largely affected by the environment. How was the influence from the environment condition change to the actuator performance?

Response: We thank the reviewer for these valuable comments. The environment condition is rather important affecting the actuation performance. For example, the oscillation actuation performance is influenced by the environment temperature. With the environmental temperature increased from 23 to 29 °C, the actuation amplitude of the PP/CB porous film increased from 0.62 to 13.46 mm under 400 mW cm^{-2} NIR light irradiation, and the actuation frequency increased from 1.54 to 4.65 Hz with the environmental temperature increased from 23 to 25 °C, and then decreased to 1.21 Hz with the environmental temperature further increased to 29 °C (Supplementary Fig. 26). (Page 11, Line 299-306)

Figure S26. Displacement, and actuation frequency of the PP/CB porous film as a function of environmental temperature under NIR light with 400 mW cm^{-2} . The size of PP/CB film is $9 \text{ mm} \times 3 \text{ mm} \times 100 \text{ }\mu\text{m}$.

7) 100 micron-thick film investigated here showed a hierarchical porous structure with microscale average pore size of 40 micron, measured by the mercury intrusion method, and a nanoscale average pore size of approximately 91.83 nm. Usually, thinner actuator may show faster actuation. However, the final blocking force will be decreased. I wonder how the specific geometries (thickness, pore size etc) were decided.

Response: Thank you for this valuable comment. We investigated the influence of the film thickness of the actuator (60, 100, and 150 μm) on the actuation stress, amplitude, and frequency. The results showed that the actuator with film thickness of 60, 100, and 150 μm exhibited the actuation stress of 26.83, 26.72, and 26.67 kPa, the amplitude of 9.98, 15.73, and 14.15 mm, and the frequency of 3.3, 3.3, and 3.3 Hz, respectively. The 100 μm thick PP/CB film almost has the best oscillation performance among the films with different thicknesses (Supplementary Fig. 24). Therefore the 100- μm -thick actuator was employed. The film pore size is also an important factor affecting the actuation performance, which would be investigated and optimized in our future work in detail. We added the discussion in the revised manuscript. (Page 10, Line 283-291)

Figure S24. Actuation stress for the porous PP/CB film (2 cm \times 1 cm) with different thickness (A), and displacement as a function of time for the oscillating bending actuation under NIR light with 800 mW cm^{-2} for the porous PP/CB film with different thickness (B). The original length and width of the film are 9 and 3 mm, respectively.

8) The photothermal actuation can be done with the monochromic (or band wavelength) light. I wonder whether the wide band or narrow band is better for efficient actuation.

Response: Thank you for this good suggestion. We added a performance test experiment using incident light with different band wavelengths (458 nm, 465 nm, 513 nm, 589 nm, and 632 nm), which exhibited actuation amplitudes of 0.54 mm, 0.25 mm, 0.35 mm, 0.24 mm, and 0.4 mm, actuation frequencies of 8.75 Hz, 7.83 Hz, 6.75 Hz, 7.83 Hz, and 7.0 Hz, and actuation stresses of 19.38 kPa, 19.42 kPa, 19.57 kPa, 18.5 kPa, and 18.15 kPa (Supplementary Fig. 37). We added this discussion in the manuscript. (Page14, Line 409-414).

Figure S37. Displacement, actuation frequency, and actuation stress as a function of incident light wavelength.

9) *The prolonged use under the intense sunlight may induce the degradation (chemical or thermal) of the polymer thin film actuator. Any long term actuation under the intense light?*

Response: Thank you for this suggestion. The oscillating displacement enhanced to 9.6 mm when the experiment was directly performed under the intense sunlight (100 mW cm⁻²) with a temperature of 36 °C, and no performance decay was observed for 20 days (Supplementary Fig. 35). We added the discussion in the revised manuscript. (Page 14, Line 394-397)

Figure S35. Displacement as a function of time for the porous PP/CB film under the intense light with a radiation temperature of about 36 °C (A), and the performance stability test for 20 days.

10) Scale bars are missing in pictures in figure 3B, 4B, 5A,D.

Response: Thank you for pointing out this important issue, and we have added the scale bars to the figures.

Response to the reviewer #3:

This submission reports the preparation of photo-responsive bending actuators comprising the polymer thin film and CB. The volume of the film could be controlled by absorbing/evaporating the solvent and this process occurred reversibly induced by the generated heat from photothermal conversion. Finally, the fabricated actuators exhibited self-oscillating behavior that is achieved under light irradiation. Whereas the technical points are well supported by the data and analysis, the originality and significance of the study are not well explained to fulfill the requirement of a high standard of the journal.

Response: We thank the reviewer for this insightful comment, and we have made a substantial revision to improve this manuscript based on the reviewers' comments.

Some specific points

1). There are more papers regarding self-oscillating photo-actuators that should be cited.

- Y. Hu, Q. Ji, M. Huang, L. Chang, C. Zhang, G. Wu, B. Zi, N. Bao, W. Chen, Y. Wu, Angew. Chem. Int. Ed. 2021, 60, 20511.

- Hu, Z., Li, Y. & Lv, Ja. Phototunable self-oscillating system driven by a self-winding fiber actuator. Nat Commun 12, 3211 (2021).

Response: We updated the literatures on self-oscillating photo-actuators and added the mentioned literatures and latest literatures in the text, as follows (Page 3, Line 67)

References:

42. Hu, Z. M., Li, Y. L. & Lv, J. A. Phototunable self-oscillating system driven by a self-winding fiber actuator. *Nat. Commun.* **12**, 3211 (2021).

43. Hu, Y. et al. Light-driven self-oscillating actuators with phototactic locomotion based on black phosphorus heterostructure. *Angew. Chem. Int. Ed.* **60**, 20511–20517 (2021).

44. Cheng, M. et al. Light-fueled polymer film capable of directional crawling, friction-controlled climbing, and self-sustained motion on a human hair. *Adv. Sci.* **9**, 2103090

(2022).

45. Zhao, T. H., Fan, Y. Y. & Lv, J. A. Photomorphogenesis of diverse autonomous traveling waves in a monolithic soft artificial muscle. *ACS Appl. Mater. Interfaces* **14**, 23839–23849 (2022).

2). *The originality of the paper is not well described. Light-driven, photo-thermal, solvent-evaporation induced, self-oscillating actuators are well known in this field. It is hard to find any scientific finding to justify the originality of the submission.*

Response: We thank the reviewer for this comment. The novelty is as follows:

Here we create a solar engine by using an oscillating actuator, which shows large oscillation displacement and high bending amplitude compared to the previous bimorph actuators. This solvent-assisted light-driven oscillator is realized by a porous film, and can respond to light irradiation from different angles.

The porous structure of the film allows fast absorption/desorption of organic solvents, and photothermal irradiation speeds up solvent evaporation and results in asymmetric film volume expansion. These combined effects lead to oscillating motions. Consequently, obvious self-oscillation behaviour occurs even under the stray light in a house.

The actuator can even oscillate while carrying a load or lifting a load under light irradiation. This provides a way to output mechanical work by directly harvesting solar energy by employing bimorph actuators. This provides the possibility of aircrafts, vehicles, and soft robotics working without the supply of fossil energy. For example, such a solar engine could be used in universe exploration, mars or moon landings, and searching, probing, or rescue in difficult-to-reach areas for human beings.

The design would inspire a wide variety of branches in machinery design, biomedicine, sensors and detectors, wearable devices and miniaturized robotics. The synergistic design of an asymmetric porous structural change assisted by solvent wetting would produce possibilities and properties when adopted for different types of functional materials. For example, a solar energy soft robot would become possible by combining actuation, sensing, and solar energy harvesting.

The current research would inspire inter-disciplinary scientific application scenarios would be developed, for example, in the fields of biomedical, flexible photovoltaic devices, electricity generation, production line with sustainable energy, and heat conduction, magnetic and acoustic applications, etc. In addition, the study on the multi-disciplinary project would produce fundamental discoveries on the mechanical, interfacial, and chemical properties.

In summary, self-oscillating actuators were achieved under divergent light, including sunlight, simulated sunlight, and infrared light, by using photothermal-induced alternating solvent evaporation from a porous PP/CB film. The actuator showed the largest oscillation displacement (15.7 mm) and amplitude (7.3 cm^{-1} or 224°) at a smaller temperature change ($5.8 \text{ }^\circ\text{C}$) compared to the oscillating actuators driven by light reported thus far. The excellent self-oscillation performance originated from the indispensable combination of the fast volume shrinkage and recovery due to the fast solvent evaporation and supply, appropriate modulus, and high flexibility of the PP/CB film, and the oscillation can never stop under continuous solvent supply. In addition, oscillating actuation can occur for the PP/CB film at different tilt angles under divergent light with different incident angles, and is applicable to a variety of volatile polar solvents, which can be used for leakage monitoring of volatile organic vapour. Moreover, oscillating actuation can still occur when carrying a load and can be used in different scenarios such as signal transmission, sensors, and controllers by replacing the loads with different functional components. The oscillating actuation can open or close a valve under light irradiation, and therefore serves as a controller to control the transmission of liquid, gas, or even light. Because the oscillating amplitude and frequency is sensitive to the light source and different types of organic solvents, it can serve as a sensor to the light intensity and wavelength by employing different types of pigments. The oscillating actuation during light transmission can be used to modulate the light signal during signal transmission.

We added the above discussion in the context to indicate the novelty and importance of this work. (Page 16, Line 451-499)

3). The authors mentioned that the fabricated photo-actuators could be operated under sunlight, however, the displacement of the actuators are too modest and slow, as shown in Figure 4b and c.

Response: We thank the reviewer for this comment. The displacement and oscillating frequency are strongly affected by the film size (length and width), mechanical properties of the actuation film, solvent types, volume expansion coefficient, the wetting height of the solvent in the film, and light intensity (Supplementary Fig. 29). In addition, the oscillating capacity would be highly affected by the pore size and pore types of the film, film mass, wavelength of the incident light, light absorption and thermal conversion capacity of the film. To improve the actuation frequency, the following factors need to be considered, including increasing the capacity of solvent absorption and evaporation, increasing the elasticity of the actuation film, and the volume change capacity during actuation. For example, we added an experiment under sunlight (100 mW cm^{-2}) at a temperature of $36 \text{ }^\circ\text{C}$, and the oscillating amplitude increased to 9.6 mm (Supplementary Fig. 35). Future work optimizing the actuation performance is being carried out. We added the discussion in the revised manuscript. (Page 12, Line 326-335) and (Page 14, Line 394-397)

Figure S29. Displacement and oscillation frequency as a function of actuating film with different lengths (A), widths (B), wetting heights of the solvent in the film (C), light intensities (D). The original size of PP/CB film is 9 mm × 3 mm × 100 μm..

Figure S35. Displacement as a function of time for the porous PP/CB film under the intense light with a radiation temperature of about 36 °C.

4). *It is not provided how much volume of the solvent is needed to operate and self-oscillate the actuators in Figure 3a. It seems that there is much amount of solvent waste even though the actuators are just waiting for the operation.*

Response: We thank the reviewer for this valuable comment. We added the following experimental details for this work. For a 0.1-mg-oscillating actuator (9 mm × 3 mm × 100 μm), 1.65 g of ethanol are used in the film and 7 g of ethanol are used in the reservoir. Because the actuator is in a sealed vessel, the evaporated ethanol can be recovered and flow back into the reservoir. Therefore, in theory almost no solvent waste occurs during such light-irradiated oscillating actuation. We added this discussion in the context. (Page 10, Line 278-283).

5). *Fig 3a. The bending oscillation of the actuators is stable and reliable over 10s. Moreover, the actuation of the first tens of cycles was not successful. Why? Is it always observed for every new session?*

Response: Thank you for this good comment. The oscillating actuation is at a resonance equilibrium state. By switching on the light irradiation, there is a time period

for the actuator starting from a static state to reach such a resonance equilibrium state (Fig. 3a). Such a time period can be shortened by optimizing the actuation conditions, such as by changing film properties, light intensity, solvent supply, etc. For example, by decreasing the ethanol supply from 7 to 1.7 g, such waiting time decreased from 17 to 1.5 s (Supplementary Fig. 21). In our future work, this point would be investigated in detail. We added this discussion in the manuscript (Page 9, Line 256-262).

Figure S21. Time as a function of ethanol mass for the PP/CB film reaching to the stable oscillation. The vertically placed porous PP film with original size of $9 \text{ mm} \times 3 \text{ mm} \times 100 \text{ }\mu\text{m}$ supplied with ethanol were used as the actuator. The vertically irradiated 800 mW cm^{-2} NIR light were used for actuation by photothermal induced solvent irradiation.

6). *What is the role of some additives (Ca, Ti)?*

Response: We thank the reviewer to point out this point. In general, the role of the inorganic additives such as CaCO_3 and TiO_2 is mainly to enhance the mechanical properties of the PP film (tensile strength, elongation at break and tensile modulus), and to improve the foam structure (cell size, cell uniformity, and cell density)^{54,55}. We added the above discussion and cited relevant literatures in the revised manuscript. (Page 4, Line 114-117)

References:

54. Chai, K. et al. Effect of nano TiO_2 on the cellular structure and mechanical

properties of wood flour/polypropylene composite foams via mold-opening foam injection molding. *J. Appl. Polym. Sci.* e52603 (2022).

55. Mohammad Mehdipour, N., Garmabi, H. & Jamalpour, S. Effect of nanosize CaCO₃ and nanoclay on morphology and properties of linear PP/branched PP blend foams. *Polym. Compos.* **40**, E227–E241 (2019).

REVIEWERS' COMMENTS

Reviewer #2 (Remarks to the Author):

The authors responded well to the comments. This should be ready for publication.

Reviewer #3 (Remarks to the Author):

The authors well improved the original manuscript to recommend acceptance of the revised submission.

Response to the reviewers:

Response to the reviewer #2:

The authors responded well to the comments. This should be ready for publication.

Response: Thanks for the reviewer contributing to our work.

Response to the reviewer #3:

Response: Thanks for the reviewer contributing to our work.